# Intrachromosomal Looping and Histone K27 Methylation Coordinately Regulates the lncRNA *H19*-Fetal Mitogen *IGF2* Imprinting Cluster in the Decidual Microenvironment of Early Pregnancy

**DOI:** 10.3390/cells11193130

**Published:** 2022-10-05

**Authors:** Xue Wen, Qi Zhang, Lei Zhou, Zhaozhi Li, Xue Wei, Wang Yang, Jiaomei Zhang, Hui Li, Zijun Xu, Xueling Cui, Songling Zhang, Yufeng Wang, Wei Li, Andrew R. Hoffman, Zhonghui Liu, Ji-Fan Hu, Jiuwei Cui

**Affiliations:** 1Key Laboratory of Organ Regeneration and Transplantation of Ministry of Education, Cancer Center, First Hospital of Jilin University, Changchun 130061, China; 2Department of Immunology, College of Basic Medical Sciences, Jilin University, Changchun 130021, China; 3Stanford University Medical School, VA Palo Alto Health Care System, Palo Alto, CA 94304, USA

**Keywords:** decidualization, recurrent spontaneous abortion, long noncoding RNA, epigenetics, H3K27 methylation

## Abstract

Recurrent spontaneous abortion (RSA) is a highly heterogeneous complication of pregnancy with the underlying mechanisms remaining uncharacterized. Dysregulated decidualization is a critical contributor to the phenotypic alterations related to pregnancy complications. To understand the molecular factors underlying RSA, we explored the role of longnoncoding RNAs (lncRNAs) in the decidual microenvironment where the crosstalk at the fetal–maternal interface occurs. By exploring RNA-seq data from RSA patients, we identified *H19*, a noncoding RNA that exhibits maternal monoallelic expression, as one of the most upregulated lncRNAs associated with RSA. The paternally expressed fetal mitogen *IGF2,* which is reciprocally coregulated with *H19* within the same imprinting cluster, was also upregulated. Notably, both genes underwent loss of imprinting, as *H19* and *IGF2* were actively transcribed from both parental alleles in some decidual tissues. This loss of imprinting in decidual tissues was associated with the loss of the H3K27m3 repressive histone marker in the *IGF2* promoter, CpG hypomethylation at the central CTCF binding site in the imprinting control center (ICR), and the loss of CTCF-mediated intrachromosomal looping. These data suggest that dysregulation of the *H19/IGF2* imprinting pathway may be an important epigenetic factor in the decidual microenvironment related to poor decidualization.

## 1. Introduction

Spontaneous abortion is the most common complication of pregnancy, affecting >20% of recognized pregnancies [1,2]. Most spontaneous abortions are sporadic and occur prior to the second trimester [3,4]. A subset of women suffer from recurrent spontaneous abortion (RSA), defined as three or more consecutive spontaneous abortions before 20 weeks of gestation. This common gynecological emergency poses significant challenges to future fertility and general psychological health.

A successful pregnancy depends upon complex crosstalk between the developmentally competent embryo and the receptive maternal endometrium [5,6]. Upon implantation, embryos elicit a complex response in the decidua, characterized by transformation of stromal fibroblasts into secretory, epithelioid-like decidual cells, accompanied by the influx of specialized uterine immune cells and vascular remodeling. Decidual cells produce growth factors and cytokines [7,8], including insulin-like growth factor binding protein 1 (*IGFBP1*) and prolactin (*PRL*), which can be used as biomarkers for decidualized cells. Abnormal endometrial receptivity is a key factor leading to implantation failure. However, the molecular factors that regulate this crosstalk in decidualization reactions remains largely uncharacterized.

Longnoncoding RNAs (lncRNAs) act as prominent epigenetic factors in normal development and numerous diseases, often by interacting with chromatin remodeling complexes [9,10,11]. Differential expression and risk analyses have identified multiple lncRNAs that are associated with recurrent miscarriage [12]. However, little is known about the specific mechanisms of these lncRNAs. Decidualization of the endometrium plays an essential role for the establishment of a successful pregnancy. In order to identify key RNA molecules that mediate the crosstalk at the fetal–maternal interface, we explored RNA transcriptome sequencing datasets from RSA patients. We found that *H19*, an imprinted lncRNA that is expressed from the maternal allele [13,14], and its reciprocally coregulated *IGF2*, a fetal mitogen gene that is expressed from the paternal allele [15,16], were highly upregulated in decidual tissues.

Genomic imprinting of the *H19*/*IGF2* cluster is regulated by the methylation status of CpG islands in the imprinting control region (ICR) located upstream of the *H19* gene. The ICR contains seven CTCF binding sites. The sixth CTCF binding site is differentially methylated [17] and serves as a CTCF “boundary insulator” [18]. Specific binding of CTCF to the unmethylated maternal allele orchestrates the formation of an intrachromosomal loop that links the *IGF2* promoters. CTCF recruits polycomb repressive complex 2 (PCR2) via the docking factor SUZ12, leading to allelic histone 3 lysine 27 (H3K27) methylation that silences the maternal *IGF2* allele. On the other hand, paternal-specific methylation of the ICR prevents CTCF binding and permits expression of *IGF2* while silencing *H19* from the paternal allele. As a result, differential methylation at the CTCF site serves as an “imprint” to ensure the reciprocal imprinting of these two neighboring genes [19]. Importantly, imprinting is dynamically regulated in gametes and in early development. Imprinting defects, including those at the *H19*/*IGF2* locus, are associated with increased risk of developmental disorders [20,21]. Aberrant DNA methylation of the CTCF binding sites in the ICR is associated with an increased risk for abortion [22] and for male infertility [23]. Furthermore, imprinting is frequently dysregulated in IVF embryos [24,25].

Given the critical role of *H19* in in vitro fertilization (IVF) [24] and male infertility [26], we examined the imprinting status of the *H19*/*IGF2* cluster in decidual tissues. We show that there is loss of imprinting of both *H19* and *IGF2* in some decidual tissues. Using human primary endometrial stromal cells as an in vitro model, we studied the epigenetic mechanisms underlying abnormal *H19/IGF2* imprinting in decidualization.

## 2. Results

### 2.1. Identification of H19 as a Recurrent Spontaneous Abortion-Associated lncRNA

To search for key factors that might be involved in fetal–maternal regulatory crosstalk in RSA, we explored the differentially expressed lncRNAs in GSE178535, which contained the RNA-seq data of decidual tissues from three RSA patients and three healthy control subjects. The Kyoto Encyclopedia of Genes and Genomes (KEGG) pathway analysis showed associations with cytokine–cytokine receptor interaction, ECM–receptor interaction, hematopoietic cell lineage, chemokine signaling pathway, *PI3K-Akt* signaling pathway, as well as signaling pathways in the regulation of stem cell pluripotency (Appendix A).

We focused on the role of the imprinted lncRNA *H19* (Figure 1A, Appendix A). In normal tissues, *H19* is expressed only from the maternal allele, while the paternal allele is silenced. Aberrant imprinting of the *H19* gene occurs frequently in tumors [19]. Using an in vitro fertilization model, we previously showed that *H19* imprinting was frequently lost in IVF embryos [24]. We were therefore interested in examining if aberrant regulation of lncRNA *H19* in decidual tissues played a role in the fetal–maternal regulatory crosstalk in RSA.

We quantitated the expression of *H19* in decidual tissues from 32 patients with RSA. For comparison, decidual tissues were also collected from 57 healthy adult women at 7–10 weeks of gestation who were undergoing early pregnancy termination (Appendix A). Using EF1A (*EEF1A1*) as the RT-qPCR control, we found that the expression of *H19* was significantly higher in decidual tissues from the patients with RSA than in decidua from healthy subjects (Figure 1B, *p* < 0.01).

The *H19* gene is located in an imprinting cluster on human chromosome 11 and is coregulated with the adjacent gene *IGF2*, a gene that encodes a mitogen that is required for normal fetal growth. The hierarchical cluster heat map analysis showed that *IGF2* was among the top six of the differentially expressed genes in the analysis (Appendix A), despite the variability among the subjects (Appendix A). Therefore, we also quantitated the mRNA abundance of *IGF2* in decidual tissues using quantitative PCR and found that, like *H19*, *IGF2* was significantly upregulated in decidual tissues derived from patients who had RSA (Figure 1C, *p* < 0.01). Similar data were also obtained by using β-Actin (*ACTB*) as the RT-qPCR control (Appendix A).

### 2.2. Loss of Genomic Imprinting in Decidual Tissues

To examine the status of *H19* and *IGF2* imprinting in decidual tissues, we genotyped genomic DNA using single nucleotide polymorphisms (SNPs) in *H19* and *IGF2*. Heterozygous SNPs were used to distinguish between the two parental alleles, and the imprinting status was examined in those tissues that were SNP-informative. Twenty-one of the decidual tissues derived from patients who had RSA were informative for *H19* heterozygosity and 20 were informative for *IGF2* heterozygosity. We found that *H19/IGF2* imprinting was lost in 39% (11/28) of *H19/IGF2* informative decidual tissues from the RSA cases (Figure 2A, left panel). Among them, 2 out of 21 samples (9.5%) showed loss of *H19* imprinting, and 7 out of 20 samples (35%) exhibited *IGF2* LOI. Two samples (#22 and #U20) showed loss of imprinting of both *H19* and *IGF2* (Table 1). Imprinting was also lost in some decidual tissues collected from the controls (Figure 2A, right panel).

As an example, the decidual tissue from Control #13 showed normal imprinting of *H19* (maintenance of imprinting) (Figure 2B, left top panel). The genomic DNA carried both the “A” and “C” alleles, but the cDNA showed the exclusive expression of the “A” allele. The “C” allele was silenced. The decidual tissues from two cases (#U18 and #M22) were also informative for the SNP (Figure 2B, left panels 2–3). However, both the “A” and “C” alleles were detected in their cDNA samples, demonstrating loss of imprinting (LOI).

Similarly, the genotyping of an SNP at the 3′-UTR of *IGF2* showed the presence of the “C/T” alleles. In normal informative decidual tissue #4, only the “T” allele was expressed (Figure 2C, top right panel). However, in two cases of RSA (U11, M22), both the “C” and “T” alleles were expressed in decidual tissues (LOI) (Figure 2C, right panels 2–3). In case U29T, however, both *H19* and *IGF2* maintained normal imprinting.

Loss of *IGF2/H19* imprinting is an early oncogenic event that is detected in tumor-paired adjacent normal tissues [19]. Therefore, we examined the allelic expression of *IGF2/H19* in decidual samples of control subjects. We also detected the presence of *IGF2/H19* LOI in the decidua of several control subjects (Appendix A), suggesting epigenetic vulnerability in the decidual microenvironment of early embryo development. The chi-squared analyses showed more LOI cases in the RSA case group for *IGF2* (*p* < 0.05, χ2= 6.93), but not for *H19* (*p* = 0.721, χ^2^= 0.407) (Appendix A). The quantitative expression data of *IGF2* and *H19* in LOI and maintenance of imprinting subgroups are presented in Appendix A. Polymorphic imprinting has been observed in placenta [27,28]. Thus, imprinting erosion as observed in both RSA and normal decidual tissues here may represent a decidua-specific polymorphic imprinting trait.

### 2.3. The Role of Altered Epigenotypes in the In Vitro Induced Decidualization Model

In vitro cell-induced decidualization is a good model for studying the complex process of implantation [29,30]. We thus cultured two human primary endometrial stromal cell lines (U29T and N45T) (Appendix A). N45T cells were cultured from the decidual tissues collected from a normal control subject. U29T cells were derived from an RSA case who had suffered four spontaneous abortions (Appendix A). Genotyping of genomic DNA showed that U29T cells were informative for both *H19* and *IGF2*. N45T cells, however, were only informative for *H19*. No informative SNPs were available for *IGF2* in N45T cells to distinguish the two parental alleles.

We examined the role of altered epigenotypes in this *in vitro* decidualization model. We pretreated U29T and N45T cells with the histone deacetylase inhibitor valproic acid (VPA) (Figure 3A), which is known to modify epigenotypes and alter allelic expression [31]. Following VPA treatment, cells were induced for decidualization. We found that this VPA treatment upregulated *IGF2* and *H19*, particularly in cells with induced decidualization (Figure 3B). However, two decidualization markers (*PRL* and *IGFBP1*) were significantly lower in VPA-treated decidualized cells (Figure 3C), suggesting an impaired decidualization process in VPA-induced cells.

### 2.4. Histone Deacetylase Inhibitor Alters Imprinting in Decidualized Cells

We then used informative SNPs to examine the allelic expression in decidualized cells (Appendix A). Both U29T and N45T cells were informative for *H19* through gDNA genotyping and both maintained normal *H19-IGF2* imprinting after being placed in culture. Maintenance of *H19* imprinting was also observed after induced decidualization, with only the “C” allele expressed in U29T cells and the “A” allele expressed in N45T cells (Appendix A, CTL-Induced cDNA). However, VPA pretreatment induced biallelic expression of *H19* in both decidualized cell lines (Appendix A, VPA-Induced cDNA). These data suggest that pretreatment with a histone deacetylase inhibitor predisposed endometrial stromal cells to lose imprinting control during decidualization.

By using informative SNP rs680, we also examined the imprinting status of *IGF2* in U29T cells (Appendix A). The untreated cells maintained normal imprinting, with only the “T” allele expressed (Appendix A, CTL cDNA). However, *IGF2* imprinting was lost, with both parental alleles (C/T) expressed in the decidualized cells (both CTL-induced and VPA-induced cDNA). *IGF2* and *H19* expression are normally tightly coordinated and reciprocally controlled by an “enhancer competition” mechanism [32]. The data from these treated primary endometrial stromal cells, however, suggest that the control of *IGF2* and *H19* imprinting can be uncoupled.

### 2.5. Loss of Imprinting Is Associated with Aberrant Histone H3 Lysine 27 Methylation

We then examined the epigenetic mechanisms underlying the loss of imprinting in these two decidualized cell lines. The expression of *IGF2* is driven by four promoters, including an upstream nonimprinted P1 promoter and three downstream imprinted promoters (P2–P4). While they are rich in CpG islands, promoters P2–P4 are not regulated by DNA methylation. Instead, gene silencing of the maternal *IGF2* allele is mediated by polycomb repressive complex 2 (PCR2) component SUZ12-catalyzed H3K27 methylation [19]. We thus focused on the status of H3K27 methylation in the three imprinted *IGF2* promoters (Figure 4A) [32].

Using antibodies specific for H3K27me3, we examined H3K27 methylation in *IGF2* promoters in U29T cells that exhibited *IGF2* LOI. We found that H3K27 methylation in the first two *IGF2*-imprinted promoters (P2, P3) was significantly reduced in decidualized cells (Figure 4B). As a control, the 5′-Ctl site upstream of the nonimprinted P1 promoter showed no significant change in the H3K27me3 mark during decidualization. In N45T cells that kept normal imprinting after in vitro decidualization, however, the ChIP signal for H3K27me3 was increased following in vitro decidualization (Figure 4C).

It is known that the key decidual marker gene *IGFBP1* in decidualization is controlled by H3K27 methylation [33]. It was therefore used as the positive control in the ChIP assay. We confirmed the reduction of H3K27 methylation in the *IGFBP1* promoter in both N45T and U29T cells following decidualization (Appendix A). As expected, decidualization did not alter the status of H3K27 methylation in the negative control gene *GPD1* (Appendix A).

### 2.6. Aberrant Imprinting Is Accompanied by the Loss of Intrachromosomal Looping

The status of histone 3 lysine 27 (H3K27) in the *IGF2* promoters is determined by CTCF-orchestrated intrachromosomal looping [34,35]. CTCF binds to unmethylated DNA motifs in ICR located between the *H19* and IGF2 genes and orchestrates the formation of an intrachromosomal loop, where polycomb repressive complex 2 (PCR2) is recruited via the docking factor SUZ12, leading to allelic H3K27 methylation which then silences the imprinted allele [36].

We used chromosome conformation capture (3C) methodology to examine the chromatin three-dimensional (3D) structure surrounding the *IGF2*/*H19* locus, with the focus on the CTCF-binding site in the ICR [37]. Using the β-Globin gene (*HBB*) as a positive control, we detected intrachromosomal looping between the LCR (locus control region) and the 3′-enhancer in two decidualized cell lines (Appendix A). In the same 3C samples, we detected an intrachromosomal loop structure between the ICR-enhancers and ICR-*IGF2* promoters in untreated U29T primary decidual cells (Figure 5A). The 3C products were purified, and DNA sequencing confirmed the loop joint separated by the Bgl2/BamH1, Bgl2/Bgl2, and BamH1/BamH1 ligation sites (Figure 5B). However, after induced decidualization in vitro, all three intrachromosomal loops were abolished (Figure 5C) in parallel with the loss of *IGF2* imprinting.

The intrachromosomal looping, however, was not significantly affected in decidualized N45T cells that maintained normal imprinting (Figure 5D). Thus, as was previously reported in cancer cells with LOI [34], CTCF-orchestrated intrachromosomal looping may be essential for maintaining normal imprinting of *IGF2* in decidual tissues.

### 2.7. Loss of Imprinting Is Associated with De Novo DNA Methylation in the Imprinting Control Region

Allelic expression of *IGF2* is regulated by the methylation status of CpG islands in the ICR. We examined allele-specific DNA methylation in the ICR for decidual tissues that were informative for two SNPs in the ICR and one SNP in the *H19* promoter (Figure 6A). The status of CpG DNA methylation was examined using sodium bisulfite sequencing. After converting the unmethylated cytosines into uracils by sodium bisulfite, the ICR and *H19* promoter regions were amplified with DNA methylation-specific primers and cloned into a pJet vector for DNA sequencing. As expected, a typical semimethylated pattern was observed in control #Z4 that had normal monoallelic expression of *H19* and *IGF2* (Appendix A). Case #M22, derived from a patient with RSA, was homozygous for the two SNPs, and therefore, we were not be able to distinguish the two parental alleles. However, we detected hyper-methylation in the ICR and the *H19* promoter (Figure 6B, top panel). Case U11, which was heterozygous for the ICR SNP, had a hyper-methylated “AA” allele and increased DNA methylation in the “AG” allele (36.5%) (left top panel).

We also observed increased CpG DNA methylation at the ICR CTCF6 site (AA allele, 19.2%) in decidualized U29T cells that exhibited *IGF2* LOI, as compared with the control cells (AA allele, 4.6%) (Appendix A). These data suggest that aberrant imprinting of *H19*/*IGF2* may be associated with CpG DNA epimutations in the ICR region.

## 3. Discussion

The molecular mechanisms underlying the spontaneous loss of a pregnancy are unknown [38]. Decidualization plays a critical role in the implantation of the embryo through a regulatory network that coordinates trophoblast invasion of the maternal decidua-myometrium and remodeling of maternal uterine spiral arteries [39,40]. Many factors, including locally secreted cytokines and growth factors, are involved in this complicated network. We have identified the lncRNA *H19* as one of the most upregulated RNA molecules in decidual tissue, where the molecular crosstalk at the fetal–maternal interface occurs. *H19* is also significantly upregulated in the decidua derived from patients with RSA. *IGF2*, a gene that encodes an important fetal mitogen, is located at the adjacent chromosomal locus. *IGF2* is also increased in the decidua in patients who have suffered an RSA. In most normal tissues, the *H19*/*IGF2* locus is imprinted. Notably, we demonstrate that there is loss of *H19* and *IGF2* imprinting in decidual tissues of some RSA patients. Loss of imprinting also occurs following induced decidualization in primary endometrial stromal cells. Mechanistically, we show that this aberrant imprinting in decidual tissues was associated with the loss of the H3K27m3 repressive histone mark as well as with the loss of intrachromosomal looping and CpG demethylation in the imprinting control center. Pretreatment with histone deacetylase inhibitor VPA predisposed primary endometrial stromal cells to develop abnormal in vitro decidualization. Collectively, these studies suggest that the disturbance of *H19/IGF2* epigenetic regulation, in addition to the locally secreted cytokines and growth factors, may be an epigenetic risk factor for poor decidualization (Figure 6C).

Both the maternal and paternal genomes are necessary for normal embryogenesis and fetal development [41,42]. *H19* is a maternally expressed imprinted gene, and its transcription gives rise to a fetal lncRNA that also functions as a precursor to microRNA miR675 [43], which negatively affects cell proliferation and tumor metastasis [44]. *H19* is abundantly expressed prior to implantation or shortly thereafter, and its expression is specifically confined to progenitor cells of the placenta and extraembryonic tissues [45,46]. *H19* is expressed coordinately with its neighboring gene *Igf2*, a gene that plays a key role in regulating fetal–placental development [47,48]. Genomic deletion of *Igf2* causes placental and fetal growth restriction. In contrast, overexpression of *Igf2* induces placental and fetal overgrowth via paracrine and/or autocrine IGF pathways. The serum levels of IGF-II have been positively linked to infant birth weight. *H19* and *Igf2* regulate embryonic development [49,50]. The allelic expression of *IGF2*/*H19* is coordinately controlled by a differentially methylated imprinting control region upstream of the *H19* promoter [19,51]. In this study, we demonstrate that both *H19* and *IGF2* are upregulated in decidual tissues of RSA patients as compared with the control cohorts. Moreover, there is loss of imprinting of both genes in many decidual tissues. Major epigenetic events take place in the embryo both in preimplantation development and in postimplantation stages, including the genome-wide resetting of imprints in the PGCs [52,53]. Aberrant methylation of imprinted genes correlates with the risk of abortion [22]. Specifically, CpG hypomethylation in the ICR is correlated with recurrent pregnancy loss [54]. As a result, the periconceptional stage is very sensitive to environmental stressors, leading to epigenetic disturbances.

Loss-of-imprinting has been linked to a number of diseases characterized by abnormal growth phenotypes and behavioral disorders, including Beckwith–Wiedemann syndrome, Silver–Russell syndrome, Angelman syndrome, and Prader–Willi syndrome [55,56], as well as multiple malignancies [57]. Placental-specific imprinting plays a critical role in coordinating the crosstalk between nutrient acquisition and fetal development. Human placentas exhibit widespread placental-specific imprints inherited from the oocyte, including maternally biased DNA methylation DMRs and histone modifications [45,50]. In particular, *H19* shows a unique placenta epigenotype, with the paternal allele-specific DNA methylation covering the core ICR to the gene body [58]. In this study, we also observed more loss of imprinting of *H19/IGF2* in RSA decidual tissues. Pretreatment of two human primary endometrial stromal cells with a histone deacetylase inhibitor induced loss of imprinting and reduced in vitro decidualization. Loss of imprinting in the placenta is associated with intrauterine growth restriction [27,59]. Future studies are needed to elucidate whether dysregulated imprinting plays a role in regulating fetal growth as well as other pregnancy-related pathologies.

It is noteworthy that the mouse and human genome contain a subset of genes that undergo polymorphic imprinting, including *IGF2*, *IGF2R*, *WT1*, *SLC22A2*, and *HTR2A*, with the imprinting status varying among individuals and tissues. For example, human *WT1* is biallelically expressed in kidney, but is monoallelically expressed in brain. In the placenta, *WT1* is maternally expressed in ~60% of the population. The human nc886 gene, encoding a tumor-suppressing ncRNA at chromosome 5q31 is another typical example of nonplacental polymorphic imprinting, with allele-specific methylation predominantly found on the maternal allele in many tissues [60]. Moreover, profiling of placental-specific imprinted DMRs shows that human placenta preferentially maintains maternal germline-derived imprint marks and appears to be highly polymorphic in the population [28]. Thus, the biallelic expression of *H19* and *IGF2* as observed in the present study may be associated with a decidua-specific polymorphic imprinting trait.

It should be noted that this study also has several weaknesses. First, two primary endometrial stromal cells yielded some discrepancies in in vitro decidualization. U29T cells, derived from the decidua of an RSA patient, were more vulnerable to hormone induction and exhibited loss of *IGF2* imprinting following in vitro decidualization. N45T cells derived from a normal subject, on the other hand, maintained normal imprinting unless they were also pretreated with histone deacetylase inhibitor. Although this discrepancy may be related to the polymorphic imprinting trait in primary endometrial stromal cells, we still do not know the specific mechanisms by which these differences arise. Second, several other lncRNAs are also upregulated in the decidual samples of RSA cases. For instance, *MALAT1* was the most upregulated lncRNA on the list. *NEAT1* was also upregulated in RSA decidual tissues. *Neat1* knockout mice stochastically show decreased fertility due to corpus luteum dysfunction and concomitant low progesterone [61], suggesting a critical role of *Neat1* in the establishment of pregnancy. Thus, future studies are needed to address if these lncRNAs are also involved in the dysregulated decidualization related to RSA.

In summary, this study reveals the first evidence that the imprinting status of *H19/IGF2* is dysregulated in decidual tissues. Using primary endometrial stromal cells as a model, we demonstrate that the in vitro decidualization process is affected by altered epigenotypes induced by a histone deacetylase inhibitor. The loss of imprinting in decidual tissues was associated with a dysregulated H3K27m3 histone marker and altered CTCF-mediated intrachromosomal looping. Altered levels of *H19* lncRNA and/or IGF-II protein in fetal decidua may alter normal fetal–placental development. It would be interesting to explore whether epigenetic targeting of the *H19*/*IGF2* epimutation [19] may provide an alternative strategy to prevent the poor decidualization seen in some pregnancy-related disorders.

## 4. Materials and Methods

### 4.1. Identification of RSA-Associated lncRNAs Using RNA-Seq Data

To identify RSA-associated lncRNAs, we downloaded the RSA dataset GSE178535 from the NIH GEO database website. The dataset contained the RNA-seq data of decidual tissues from three RSA patients and three healthy control subjects [62] (Next Generation Sequencing Facilitates Quantitative Analysis of healthy controls and RSA patients Transcriptomes. Available online: https://www.ncbi.nlm.nih.gov/geo/query/acc.cgi?acc=GSE178535, accessed on 22 June 2021).

The in vitro decidualization of embryonic stem cells (ESCs) was induced using differentiation media containing 0.5 mM dibutyryl cAMP, 1 µM medroxyprogesterone 17-acetate, and 10 nM β-estradiol. Decidualized cells were used for RNA-seq [63].

Differentially expressed RNAs were calculated as the log2-transformed gene expression values (Fold Change). The Kyoto Encyclopedia of Genes and Genomes (KEGG) pathway analysis (KEGG_PATHWAY) was carried out using DAVID Bioinformatics Resources 6.8 (https://david.ncifcrf.gov, accessed on 21 September 2022) [64,65]. Hierarchical Cluster Heatmap was generated using HIPLOT (https://hiplot.com.cn, accessed on 21 September 2022) [66]. LncRNAs with the fold-change >2 and *p* < 0.001 were chosen for further functional characterization.

### 4.2. Human Decidual Samples

Decidual tissue samples were collected at The First Hospital of Jilin University between 2017–2022. Ethical approval for this study was provided by the Research Ethics Board of the First Hospital of Jilin University, and written informed consent was obtained from all patients prior to sample collection.

A total of 32 decidual tissues were collected from women with unexplained RSA. The inclusion criteria for this group were women aged under 40 years with a history of > three consecutive pregnancy losses. Clinical examination showed that they had normal uterine cavity shape and size; normal follicle-stimulating hormone (FSH), estradiol (E2), prolactin (PRL), luteinizing hormone (LH), and thyroid-stimulating hormone (TSH) levels at menstrual day 2–3; no mutations detected in Factor V (Leiden) and prothrombin gene analysis; normal antithrombin III, protein C, and S activity; negative results for lupus anticoagulant evaluation; cardiolipin antibody; beta2-glycoprotein antibody; and normal karyotype. Their partners have normal semen analyses and normal karyotype. None of the patients had received a prior infertility treatment.

In addition, 57 decidual samples were obtained as the control group from healthy adult women at 7–10 weeks of gestation undergoing legal elective termination. The inclusion criteria were women aged under 40 years with regular menstrual cycles, at least one live birth, no previous miscarriages, no history of infertility/treatment, and no associated gynecologic (endometriosis, fibroids, active or history of pelvic inflammatory disease) or other medical comorbidities (e.g., hyperprolactinemia, thyroid disease). The male partners of control subjects had normal semen analysis results and karyotypes. The characteristics of RSA patients and controls are listed in Appendix A.

All the decidual samples were collected by the same pathology lab technician at Jilin University First Hospital. The placenta was rinsed with saline to remove blood. Decidual tissues were collected by carefully dissecting the maternal basal plate of the placenta. Collected tissues were rinsed with 1× PBS, frozen with liquid nitrogen, and saved in –80 °C freezer for analysis.

### 4.3. Culture of Human Primary Endometrial Stromal Cells

Primary endometrial stromal cells were cultured from U29T and N45T decidual tissues that were *H19-IGF2* informative and that maintained normal imprinting. N45T cells were cultured from the decidual tissues collected from a normal control subject. U29T cells were derived from an RSA case who had suffered four spontaneous abortions (Appendix A).

After curettage, the tissues were immediately collected under sterile conditions into prechilled PBS and divided into decidua and villi. Two or three pieces of decidual tissues were collected and washed 2–3 times again with prechilled PBS to exclude villous contamination. Fresh tissues were cut into approximately 2 mm^3^ fragments, washed in DMEM (high glucose; Sigma, MO, USA), and directly cultured at 37 °C in 5% CO_2_ by attaching to the substratum in a 10 cm dish with complete medium consisting of DMEM medium (Sigma, St. Louis, MO, USA) supplemented with 10% (*v*/*v*) fetal bovine serum (Sigma, St. Louis, MO, USA), 100 U/mL of penicillin sodium, and 100 µg/mL of streptomycin sulfate (Invitrogen, Carlsbad, CA, USA). After approximately 12 days in culture, cells migrated out from the edges. Migrating cells were collected with 0.1% trypsin and 0.25 mM EDTA and passaged for allelic study and in vitro decidualization assays (Appendix A). After culturing, cells were aliquoted and stored in liquid nitrogen for further studies.

### 4.4. In Vitro Decidualization

In vitro artificially induced decidualization was performed following the method as described in [29]. Briefly, U29T and N45T primary endometrial stromal cells were cultured in complete medium containing 10 nM E2, 1 µM P4, and 0.5 mM 8-Br-cAMP. Culture medium was changed every 2 days. Cells were harvested for subsequent experiments 96 h after the treatment.

### 4.5. The Role of Histone Deacetylase Inhibitor VPA in Decidualization

To examine the role of aberrant epigenotypes in in vitro decidualization, we pretreated primary endometrial stromal cells with the histone deacetylase inhibitor valproic acid (VPA), which is known to modify epigenotypes and alter allelic expression [31]. U29T and N45T cells were treated with 2 mM VPA. Cells treated with equal volume of PBS were used as the control (Ct). Culture medium was changed daily. Forty-eight hours after VPA treatment, cells were used for in vitro decidualization experiments. After 96-h treatment, cells were collected for imprinting assays.

### 4.6. RT-PCR Quantitation

Decidual tissues and cells were collected and total RNA was extracted by TRIzol reagent (Sigma, St. Louis, MO, USA) and stored at −80 °C. cDNA was synthesized using RNA reverse transcriptase (Invitrogen, CA, USA), and target amplification was performed with a Bio-Rad Thermol Cycler. PCR of 1 cycle at 95 °C for 2 min; 32 cycles at 95 °C for 15 s, 60 °C for 15 s, and 72 °C for 15 s; and 1 cycle at 72 °C for 10 min. EF1A (*EEF1A1*) and β-Actin (*ACTB*) were used as the internal controls. Quantitative real-time PCR was performed using SYBR GREEN PCR Master (Applied Biosystems, Foster City, CA, USA); the threshold cycle (Ct) values of target genes were assessed by quantitative PCR in triplicate using a sequence detector (ABI Prism 7900HT; Applied Biosystems, Foster City, CA, USA) and were normalized over the Ct of the EF1A or β-Actin controls. Primers used for PCR quantitation are listed in Appendix A.

### 4.7. Allelic Expression of IGF2 and H19

Genomic DNA and total RNA extraction from decidual tissues and cDNA synthesis were performed as previously described. Decidual tissues were first genotyped for heterozygosity of SNPs in *IGF2* exon 9 and *H19* exon 5 (Figure 2A). Target amplification was performed with a Bio-Rad Thermol Cycler. PCR of 1 cycle at 95 °C for 2 min; 32 cycles at 95 °C for 15 s, 60 °C for 15 s, and 72 °C for 15 s; and 1 cycle at 72 °C for 10 min using primers specific for two polymorphic restriction enzymes (ApaI, AluI) in the last exon of human *IGF2* and *H19* exon 5. To determine the status of *IGF2* imprinting, the amplified products were sequenced by Comate Bioscience Co, Ltd. (Changchun, China). Decidual tissues that maintain normal imprinting expressed a single parental allele, while the LOI showed biallelic expression of *IGF2* and *H19*. PCR primers used for *IGF2* imprinting are listed in Appendix A.

### 4.8. DNA Methylation Analysis

Genomic DNA was collected from tissues or cells using dBIOZOL Genomic DNA Extraction Reagent (BioFlux, BSC16M1, Hangzhou, China) following the manufacturer’s instructions. DNA was treated with EZ DNA Methylation-Gold^TM^ Kit (ZYMO RESEARCH, D5005, Irvine, CA, USA), and PCR was performed using DNA methylation-specific primers designed for the promoter of *H19* and CTCF binding sites (Appendix A). To examine the status of DNA methylation in every CpG site, the amplified PCR DNAs were cloned into pJET1.2/blunt cloning vector (Thermo, K1231, Waltham, MA, USA) and transformed into TOP10. Plasmid DNA was collected by Wizard^®^ Plasmid DNA Purification kit (Promega, A1223, MO, USA) and sequenced.

### 4.9. Chromosome Conformation Capture (3C)

Furthermore, 3C assays were performed to determine long-range intrachromosomal interactions as previously described [35,67,68,69]. Briefly, 1.0 × 10^7^ cells were cross-linked with 2% formaldehyde and lysed with cell lysis buffer (10 mM Tris (pH 8.0), 10 mM NaCl, 0.2% NP-40, supplemented with protease inhibitors). Nuclei were collected and suspended in 1× restriction enzyme buffer. An aliquot of nuclei (2 × 10^6^) was digested with 800 U of restriction enzyme BamH1/Bgl2 at 37 °C overnight. After stopping the reaction by adding 1.6% SDS and incubating the mixture at 65 °C for 20 min, chromatin DNA was diluted with NEB ligation reaction buffer, and 2 μg DNA was ligated with 4000 U of T4 DNA ligase (New England BioLabs, Irvine, CA, USA) at 16 °C for 4 h (final DNA concentration, 2.5 μg/mL). After treatment with 10mg/mL proteinase K at 65 °C for 4h to reverse cross-links and with 0.4 μg/mL RNase A for 30 min at 37 °C, DNA was extracted with phenol-chloroform, ethanol precipitated, and detected by PCR amplification of the ligated DNA products. Furthermore, 3C PCR products were cloned and sequenced to validate the intrachromosomal interactions by assessing for the presence of the BamH I/Bgl II ligation site. The 3C interaction was quantitated by qPCR and was standardized over the 3C ligation control. For comparison, the relative 3C interaction was calculated by setting the control as 1.

As the 3C quality control, human β-Globin (*HBB*) gene was used as a positive control. Unlike *IGF2* promoters P2-P4, *IGF2* promoter 1 (P1) is not imprinted and is biallelically expressed in all tissues. Thus, we chose a Bgl2 site upstream of P1 promoter as the 3C negative control. Human Primers used for 3C assay are listed in Appendix A.

### 4.10. Histone Methylation by Chromatin Immunoprecipitation (ChIP) Assay

As previously described, a ChIP assay was used to quantitate the status of histone modifications following the manufacturer’s protocol (Upstate Biotechnology, Lake Placid, NY, USA). Briefly, 1.0 × 10^7^ cells were fixed with 1% formaldehyde and then sonicated for 180 s (10 s on and 10 s off) on ice with a sonicator with a 2mm microtip at 40% output control and 90% duty cycle settings. The sonicated chromatin was collected by centrifugation, aliquoted, and stored at −80 °C. Protein A/G Magnetic Beads and a specific anti-trimethyl-histone H3 (Lys27) antibody (Merck Millipore, Darmstadt, Germany) were incubated with rotation for 30 min at room temperature. The sonication supernatant and beads were incubated with antibody at 4 °C on a rotating rack for 4–16 h or overnight. To reduce the ChIP background, we modified the manufacturer’s protocol by adding two more washing steps following immunoprecipitation. As previously reported [35], anti-IgG was used as the ChIP control in parallel with testing samples. Precipitated DNA was subjected to qPCR and expressed as fold-enrichment compared to the IgG chromatin input.

For the ChIP assay, *IGFBP1*, a key decidual marker gene controlled by H3K27 methylation in decidualization, was used as the positive control. The housekeeping gene *GPD1* (G3PDH) was used as the negative control in the assay.

### 4.11. Statistical Analysis

All the experimental data are presented as mean ± standard deviation (SD) and were derived from at least three biological replicates. Statistical analyses were performed using GraphPad Prism v7.0 (GraphPad Software, San Diego, CA, USA). Unpaired two-tailed Student’s *t*-tests were used for comparison between two groups. One-way ANOVA with Bonferroni’s multiple comparison test was used to compare statistical differences for variables among three or more groups. Chi-squared tests were used to examine the association between the *H19/IGF2* imprinting status (loss of imprinting and maintenance groups) and the risk of RSA occurrence (RSA and control groups). The level of significance was indicated as * *p* < 0.05, ** *p* < 0.01, and *** *p* < 0.001, unless stated otherwise.

## Figures and Tables

**Figure 1 cells-11-03130-f001:**
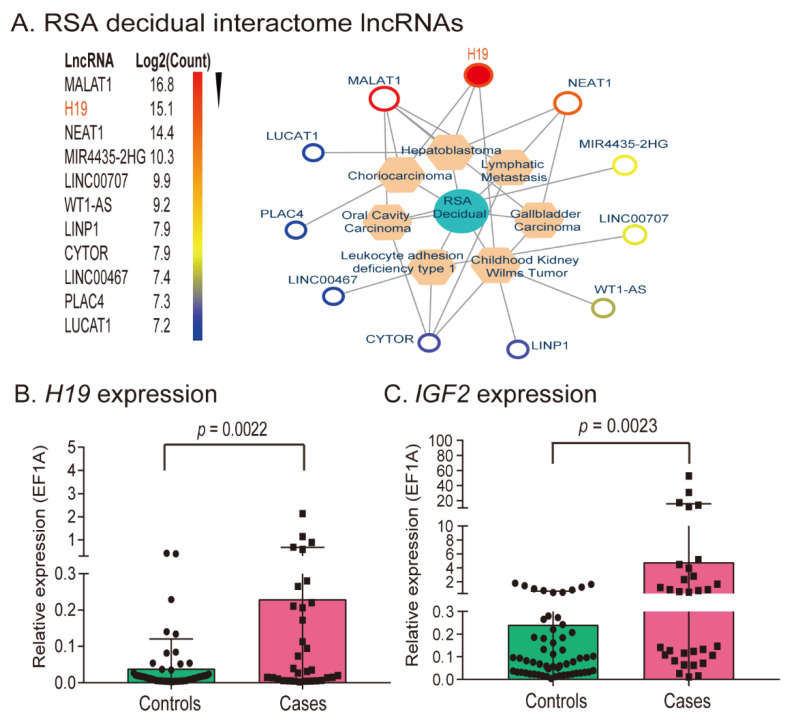
Identification of RSA-associated lncRNAs by integrating RNA-seq data from RSA patients and senescent decidualized human endometrial stromal cells. (**A**) Identification of RSA-associated lncRNAs. Differentially expressed lncRNAs were analyzed using the RNA transcriptome sequencing dataset GSE178535. The top 11 differentially expressed lncRNAs are ranked based on the RNA expression-fold from high (red) to low (blue) between the RSA patients and the controls. (**B**) Upregulated *H19* in decidual tissues from RSA cases. Thirty-two decidual tissues with unexplained RSA were collected as the case group. As the control, 57 decidual tissues samples were obtained from healthy adult women who were diagnosed with early pregnancy and were undergoing legal elective termination. Gene expression was measured by qPCR and standardized over the value of the *EEF1A1* control. All data shown are mean ± SD. Error bars represent the SD of the average of three independent PCR reactions. *p* = 0.0022 as compared with the CTL control. (**C**) Reciprocal upregulation of *IGF2* in decidual tissues of RSA cases. *p* = 0.0023 as compared with the CTL control.

**Figure 2 cells-11-03130-f002:**
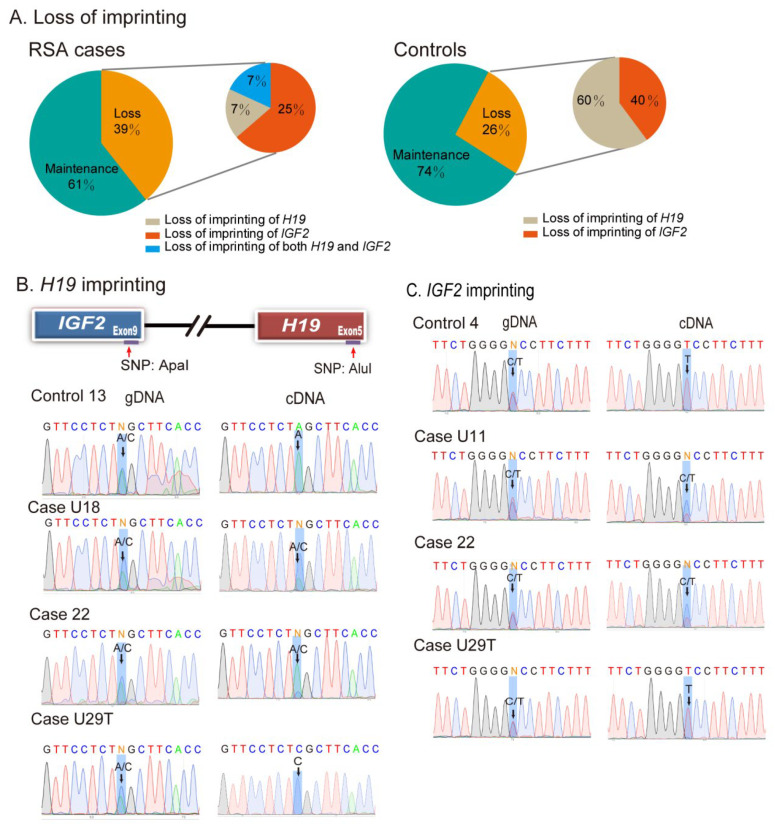
Loss of *H19*/*IGF2* imprinting in decidual tissues in RSA. (**A**) Percentage of abnormal *H19*/*IGF2* imprinting. Among the decidual tissues that are Apa1-informative, 39% cases in RSA cases and 26% in control subjects show the loss of either *H19* or/and *IGF2* imprinting. (**B**) Example of aberrant *H19* allelic expression in RSA cases. Genomic DNAs (gDNA) of both cases U18 and M22 are Apa1 SNP informative (A/C alleles). In the cDNA samples, both parental alleles are expressed in decidual tissues. (**C**) Loss of *IGF2* imprinting in RSA. In control 4, only the T allele is expressed. However, in cases U11 and M22, decidual cDNAs show biallelic expression of *IGF2*.

**Figure 3 cells-11-03130-f003:**
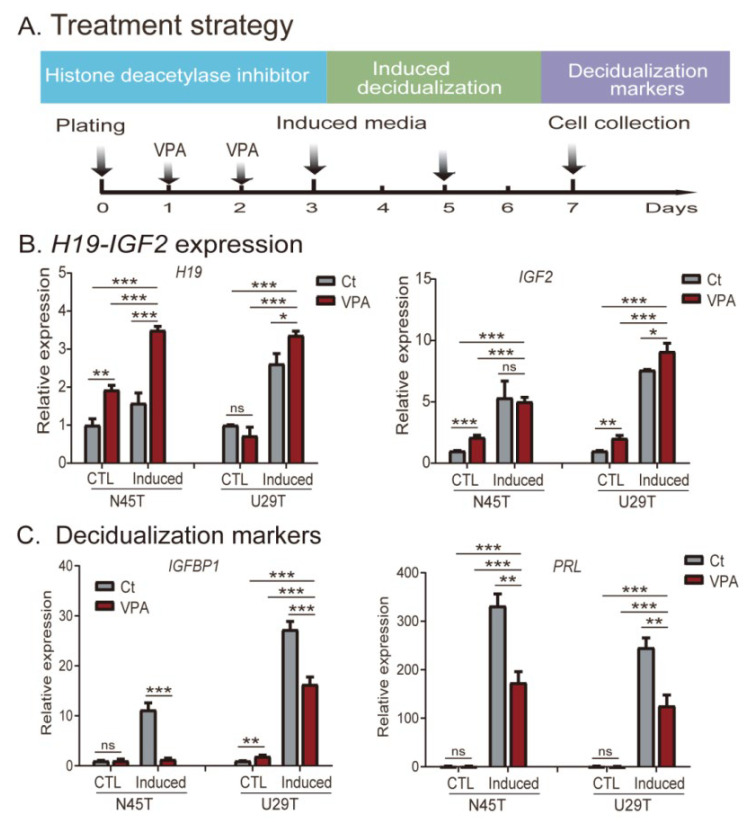
The role of disturbed epigenetics in in vitro decidualization. (**A**) Strategy of inducing epigenetic disturbance by the histone deacetylase inhibitor VPA in primary cultures of endometrial cells. Cells were pretreated with VPA and then were induced for in vitro decidualization. (**B**) Expression of *H19* and *IGF2* in decidualized endometrial cells. VPA: cells treated with the histone deacetylase inhibitor valproic acid. Ct: cells treated with PBS control. Induced: in vitro induction of decidualization. CTL: PBS-treated control cells. (**C**) Expression of two decidualization markers *IGFBP1* and *PRL* in decidualized endometrial cells. The data are the mean ± SD from three independent experiments. * *p* < 0.05, ** *p* < 0.01, and *** *p* < 0.001 as compared with the vector lentivirus control group (CTL). ns: not statistically significant.

**Figure 4 cells-11-03130-f004:**
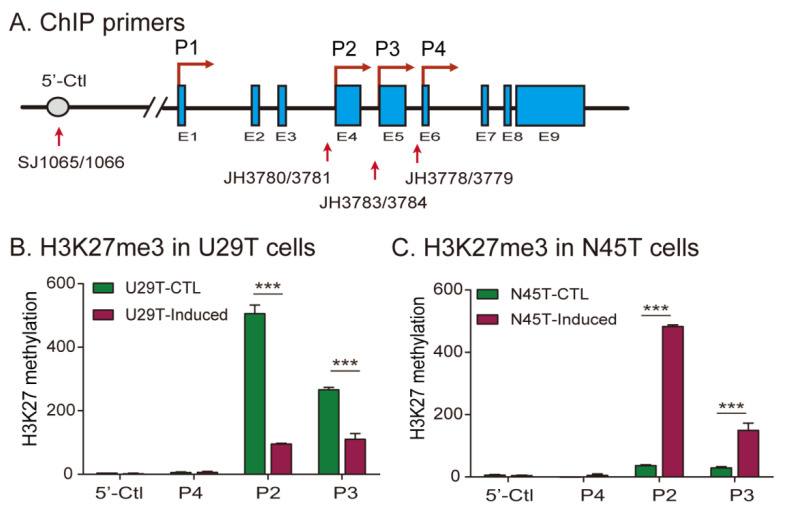
H3K27 methylation in the promoter of *IGF2*. (**A**) Location of PCR primers used for H3K27 methylation. Two primer sets JH3780/JH3781 and JH3783/JH3784 are used to quantitate H3K27 methylation in promoters 2 and 3 of *IGF2*. The primer set (SJ1065/SJ1066) for the P1 promoter upstream site (5′-Ctl) is used as the negative control. (**B**,**C**) Histone methylation in the *IGF2* promoter of U29T cells (**B**) and N45T (**C**) cell lines. U29T cells exhibited *IGF2* LOI after decidualization, while N45T cells maintained normal imprinting. Histone modifications in the *IGF2* promoter were measured by ChIP assay using antibodies specific for H3K27me3. Normal rabbit IgG was used as a negative control and was used for normalization. The data are the mean ± SD from three independent experiments. *** *p* < 0.001as compared with control cells (CTL). Note the reduced H3K27me3 level in decidualized N29T cells that demonstrate *IGF2* LOI.

**Figure 5 cells-11-03130-f005:**
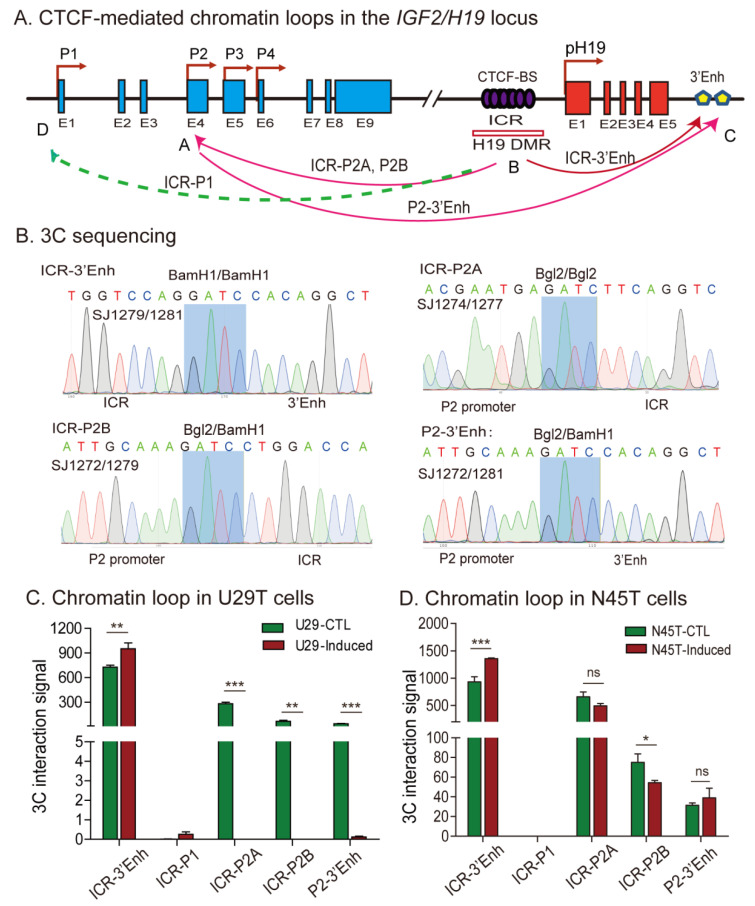
Intrachromosomal loop interactions in the *H19*/*IGF2* imprinting locus. (**A**) Location of 3C primers used to detect the interaction between the *IGF2* promoter, CTCT6, and *H19* enhancer. P1–P4: *IGF2* promoters. E1–E9: *IGF2* exons 1–9. E1–E5: *H19* exons 1–5. Enh: enhancers. Arrows: intrachromosomal interactions. (**B**) Sequencing of the *IGF2*/*H19* intrachromosomal loop 3C products. Blue background on the sequence: the 3C ligation product between the BamH1 and/or Bgl2 restriction sites. (**C**,**D**) Quantitation of 3C intrachromosomal interaction signals. The 3C interaction was quantitated by qPCR. U29T cells showed normal imprinting but exhibited *IGF2* LOI after decidualization. Note the lack of intrachromosomal interaction in decidualized N29T cells. Decidualized N45T cells with normal imprinting were used as the control. The data represent the mean ± SD from three independent experiments. * *p* < 0.05, ** *p* < 0.01, and *** *p* < 0.001 as compared with the vector lentivirus control group (CTL). ns: Not statistically significant.

**Figure 6 cells-11-03130-f006:**
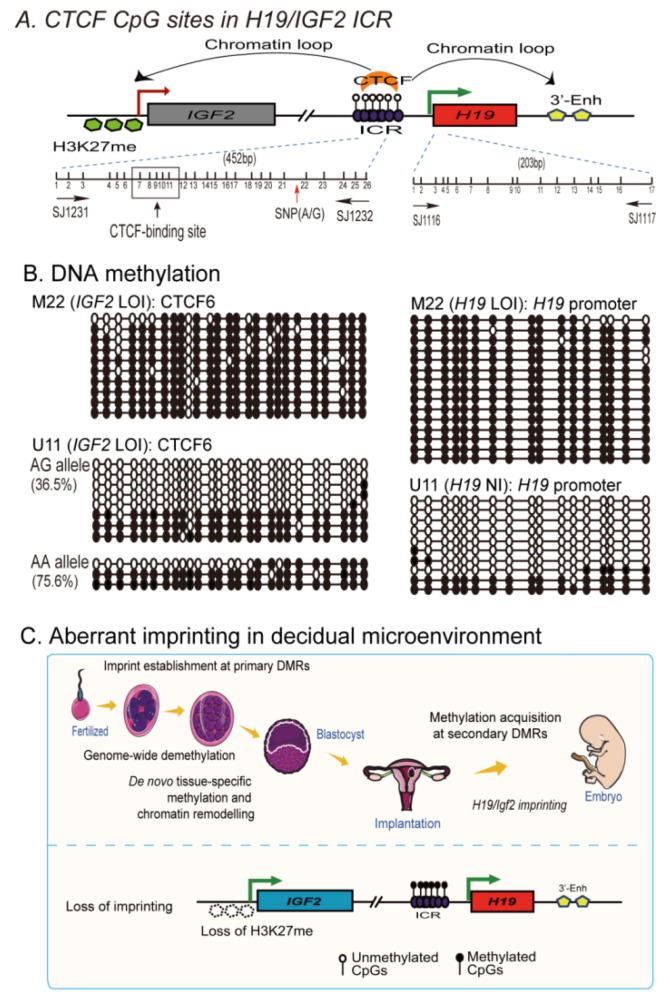
Abnormal DNA methylation at the CTCF6 site in the imprinting control region. (**A**) Schematic diagram of CpGs in the ICR. Locations of PCR primers are indicated by numbered arrows. Vertical lines: location of CpGs. Red arrows: single nucleotide polymorphisms allowing for discrimination of the two parental alleles in case U11. CTCF site 6 carrying CpG 7–11 is boxed. CTCF, a tethering protein, binds to the unmethylated ICR and serves as a molecular glue to secure intrachromosomal interactions. The CTCF-mediated looping brings the ICR and the *IGF2* promoters into close contact, where the polycomb repressive complex 2 (PCR2) is recruited via SUZ12, inducing allelic H3K27 methylation and epigenetic imprinting. (**B**) Alteration of DNA methylation at the CTCF6 site and the *H19* promoter in decidual tissues of cases with RSA (M22, U11). LOI: loss of imprinting. NI: not informative. Numbers in parenthesis: percentage of methylated CpGs. Note the hypermethylation status in case M22 and biallelic DNA methylation in case U11 at the CTCF6 site. (**C**) A model of aberrant imprinting in decidual tissues. After fertilization, genome-wide demethylation occurs, except for DMRs. Following embryo implantation, a global de novo methylation occurs in response to organ development. Parental-specific DMR imprints determine tissue-specific allelic expression of imprinting genes, including *H19/IGF2*. Loss of *H19/IGF2* imprinting occurs in the decidual microenvironment due to the aberrant control of the ICR epigenotype, intrachromosomal looping, and H3K27m3 repressive histone marks. Dysregulation of *H19/IGF2* in preimplantation development and postimplantation stages may represent an epigenetic risk factor contributing to abnormal decidual microenvironment, in addition to locally secreted cytokines and growth factors.

**Table 1 cells-11-03130-t001:** Loss of *H19* and *IGF2* imprinting in RSA decidua.

		*H19*		*IGF2*	
Cases (ID)		Genotype	cDNA	Genotype	cDNA
Loss of imprinting of *H19* (9.5%) *					
1	U18	A/C	a/c	C/T	t
2	U21	A/C	a/c	T/T	-
Loss of imprinting of *IGF2* (35%) **					
1	8	A/C	a	C/T	c/t
2	E1	A/C	c	C/T	c/t
3	E3	A/A	-	C/T	c/t
4	E5	A/C	c	C/T	c/t
5	U11	A/A	-	C/T	c/t
6	U14	A/A	-	C/T	c/t
7	U17	A/A	-	C/T	c/t
Loss of imprinting of *H19* and *IGF2 ****					
1	M22	A/C	a/c	C/T	c/t
2	U20	A/C	a/c	C/T	c/t

* After genotyping, 21 informative samples were used for *H19* allelic analysis. ** 20 *IGF2*-informative samples were used to examine *IGF2* imprinting. *** Informative for both *H19* and *IGF2.* - Tissues that are not informative for allelic analysis of either *H19* or *IGF2.*

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
