# Peer review of "Intrachromosomal Looping and Histone K27 Methylation Coordinately Regulates the lncRNA H19-Fetal Mitogen IGF2 Imprinting Cluster in the Decidual Microenvironment of Early Pregnancy"

_cells, 2022, doi:10.3390/cells11193130_

Round 1
Reviewer 1 Report
This is a very well-written manuscript of high quality and of an actual and interesting topic. The experimental design is well performed.
In my opinion the manuscript report novel relevant genetic, epigenetic and functional data on the role of lncRNA H19 in decidualization during early pregnancy. The work is in agreement with the scope of the Journal.
Several, major comments on my side are for improving the manuscript:
1. Several typo errors are present. For instance lines 23, 58, 59, 96 etc. other typos are present throughout the text. Authors are encouraged to carefully revise the manuscript in order to fix these typo
2. The aim of the work at the end of the introduction should be more deeply described. iT will improve the reading and quality of the work
3. General minor comment. The periods after the panel annotations, in all figure captions are unnecessary and can be removed, e.g., line 136, “A).”, line 138 “B).”.
4. Both strengths and weaknesses of the study should be included in the discussion
5. A conclusive short paragraph describing the main findings and observations should be included at the end of the discussion
minor observations:
line 23 Please revise this typo error
line 54 considering the topic, a couple of sentences of the current knowledge on lncRNAs dyrtegulation and RSA sohuld be included as a background. For instance PMID: 35397764
Lines 63-79 a detailed description of the structure and function H19/IGF2 cluster, alongside its epigenetic regulation and its implication in male infertility and early pregnancy loss is also reported here ( DOI: 10.3389/fcell.2021.689624 ). For completeness, this important reference should be included.
Line 71 and or 93 The CTCF binding has also been found differentially methylated in sperm DNA from infertile males PMID: 23975186. In other words, aberrant Imprinting of H19 has been related to male infertility
The quality of chromatograms in figures 2 and 5 panel B in both cases should be improved as being difficult to read
Lines 202-208 This section should be reorganized with a short subhead title (without figure citation) and several following sentences
Line 209 I suggest that H3K27 as well as other acronyms (if present) should be mentioned with their complete names in the subhead titles
Line 304 “de novo” as well as other Latinisms should be in italic style when mentioned. Please revise the entire text accordingly
Line 320 I discourage the use of underlined words in a scientific article. Similarly, figure citations should be avoided in the discussion
Line 340 Better “microRNA miR675”
Line 410 ESC cells should be mentioned with their complete name the first time being mentioned
Sperogramm information of the male partners of control females should be mentioned. Was it presenting normal parameters?
Reviewer 2 Report
IGF2 is known to be involved in embryo mitogen, and this study presents a hypothesis and its validation data that its abnormal expression occurs via abnormal imprinting of H19, an lncRNA, and causes RSA, which is very interesting and important data. But it still needs a considerable revision to be acceptable for “Cells”.
<Major concerns>
IGF2 is high in CTL1 and CTL3 and low in RSA3. It would be good to look properly at individual data as well as differences in expression (Line88, Fig. S1). In fact, looking at the FIG1BC, it appears that there is no difference between controls and RSA with most of them having low values, and in H19, there are scattered samples that are high in both groups. In addition, since the statistical analysis for Figure 1BC is questionable, the expression data for all 89 specimens should be submitted as supplement data. In addition, authors should also compare between IGF2 expression data and that of the imprinting status. Furthermore, it is known that β-Actin (Line 484) cannot be used as an internal standard because of its large variability in endometrial tissues. Elongation factor-1α would be a suitable alternative, and the expression data should be re-produced using EF1A.
Line 170: If Primary cells, authors should clarify the patient information of origin. Also If these are Primary cells, authors have only seen two cases, which raises concerns about the reliability of the data. If these are cell lines, authors should indicate the sources and it raises the concern that the data may not reflect normality. Even though H19 is elevated, IGF2 is not elevated in N45T (Line175). Also, there is no significant change in N45T on Line272. These are very important findings, and I wonder if the difference in origin between N45T and U29T may be quite relevant. In that sense, the information on N45T and U29T is very important, and I'm wondering if it is necessary to discuss not only the differences between them, but also the mechanisms by which these differences arise.
Line 544: Insufficient information is available.
<Minor comments>
Line 40: “may” should be deleted.
Line 90: Isn't most upregulated gene MALAT1?
Line107-108: Are the correct error bars SD or SE?
Line 118: We can see some variation within the group, so maybe it is just being pulled by the unusually high numbers?
Round 2
Reviewer 2 Report
In this revised version, the authors have improved the manuscript appropriately in response to my comments.
I no longer have any comments and I would recommend this version of the review for publication.